# Research on the Anti-Disturbance Control Method of Brake-by-Wire Unit for Electric Vehicles

**Xiaoxiang Gong [1,]*, Lixia Qian [1], Weiguo Ge [2] and Lifeng Wang [2]**

1   Chongqing Engineering Research Center for Advanced Intelligent Manufacturing Technology, Chongqing
    Three Gorges University, Chongqing 404000, China; 20160014@sanxiau.edu.cn
2   Chongqing Engineering Technology Research Center for Light Alloy and Processing, Chongqing Three
    Gorges University, Chongqing 404000, China; 13668422133@163.com (W.G.); lifengcumt@163.com (L.W.)
*   Correspondence: gongxiaoxiang@126.com; Tel.: +86-152-0231-0462

**Abstract:** In order to improve the braking performance of electric vehicles, a novel brake-by-wire actuator based on an electro-magnetic linear motor was designed and manufactured. For the purpose of braking force regulation accuracy and high robust performance, the state observer and the anti-disturbance controller were designed in this paper after describing the actuator structure, braking principle, and mathematical model. The simulation and experimental results showed that the brake actuator responded rapidly, since its response time was only 15 ms. Compared to traditional PID (Proportion Integration Differentiation) methods, the controller proposed in this paper is able to regulate the braking force more precisely and has better anti-disturbance performance, thus the braking process can be accurately controlled according to the driver's demand. The vehicle simulation results showed that the braking distance and braking time were shortened by 12.19% and 15.54%, respectively compared with those of the conventional anti-lock brake system (ABS) in the same braking conditions.

**Keywords:** electro-magnetic linear actuator; brake-by-wire; anti-disturbance; state observer; control performance

---

## 1. Introduction

Vehicle braking technology continuous to improve due to the increasing concern about safety performance. Active safety systems, such as anti-lock brake system (ABS), electronic brake force distribution system (EBD), traction control system (TCS), yaw stability control system (YSC), and electronic stability program system (ESP) were developed successively in recent decades. In particular, the ABS has become the prerequisite equipment for passenger cars, and a large number of researchers are constantly improving its performance [1–3]. While greatly improving, the brake system is also becoming more complex and more difficult. More importantly, it still cannot regulate the braking force independently and accurately between wheels according to the electrical vehicle (EV)'s regenerative braking strategy, therefore it is not adapting to the development of EV [4]. The brake-by-wire (BBW) systems, represented by electro-mechanical brake (EMB), electro-wedge brake (EWB), and electronic hydraulic brake (EHB), are significant improvements introduced since the ABS was developed [5–7]. Because the BBW systems are able not only to improve the braking performance effectively but also to adapt well to the regenerative braking system, the kinetic energy can be significantly recovered in EV braking [8].

The mechanical/hydraulic connections between pedal and wheel actuator are replaced by signal and power cable in the BBW system, and the wheel actuators are driven independently by motors. Thus, the braking force of every wheel can be independently adjusted and changed, and the braking

process can be regulated more flexibly than with a hydraulic system [9]. At present, many researchers are studying how to optimize the structure of the BBW actuator and to improve the performance of the controller. Peng focused on the EMB based on brushless direct current motor and designed a fuzzy sliding mode controller for torque regulation [10]. Han designed a robust sliding mode controller to improve the EWB performance [11]. Lee proposed a novel active brake judder attenuation strategy for EMB and designed two attenuation algorithms to compensate the clamp force so as to eliminate the judder which caused brake torque variations [12]. In addition, researchers such as Todeschini, Shin, Lindvai, and Wang have proposed their own methods for the optimization design of the BBW actuator and the improvement of the controller [13–16].

In the BBW solutions that have been proposed so far, both EMB and EWB are driven by rotary motors. First, the motor torque must be enlarged by a gear mechanism and then be converted into linear force by a ball-screw mechanism in the EMB. Similarly, the torque must be converted into linear force by a ball-screw mechanism first and then the force must be enlarged using a wedge mechanism in the EWB. Therefore, the structure of EMB and EWB are complex because they require both motion conversion and torque/force amplification mechanisms at the same time [11,15]. On the other hand, the EHB was improved with respect to the conventional hydraulic braking system by adding extra solenoid valves. However, it still has a hydraulic pump, hydraulic pipelines, hydraulic valves, and other hydraulic components, and the braking force of all wheels are still powered by the same hydraulic pump, thus its braking performance is only marginally improved [17].

In order to effectively deal with the complex structure of EMB and EWB and further improve the braking performance, a novel BBW actuator based on an electro-magnetic linear actuator (EMLA) was designed in this work and was named direct-drive electro-hydraulic brake (DDEHB). Then, the state observer and the anti-disturbance controller (ADC) were designed to improve response time and control precision thus achieving a robust performance of the DDEHB. Afterwards, the DDEHB characteristics and control strategy were tested by simulation. Finally, a co-simulation model based on MATLAB/Simulink (R2014a) and AMESim (Rev 13) was built to validate the vehicle braking performance.

## 2. Structure and Model of the DDEHB

### 2.1. Structure and Principle of the DDEHB

The DDEHB is a novel brake-by-wire actuator that utilizes a fast-responding EMLA to provide brake force. The EMLA converts electrical energy into electro-magnetic force, and then the force is amplified by an unequal-diameter hydraulic cylinder (UDHC) before being forced on the brake disc. The structure of the DDEHB is shown in Figure 1 [18].

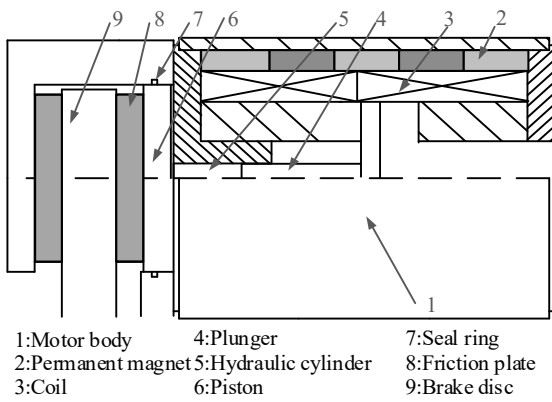

| | | |
|---|---|---|
| 1:Motor body | 4:Plunger | 7:Seal ring |
| 2:Permanent magnet | 5:Hydraulic cylinder | 8:Friction plate |
| 3:Coil | 6:Piston | 9:Brake disc |

**Figure 1.** Structure of the direct-drive electro-hydraulic brake (DDEHB).

Figure 1 shows the EMLA body. The permanent magnets *2* are pasted on the body's inner surface. The coil *3* moves along the axis when it is energized by the electromagnetic force. The plunger 4 is connected with a coil by a pin and moves together with the coil. The motion of the plunger (direction, velocity, and displacement) depends on the magnitude and direction of the coil current. Two friction plates 8 are symmetrically installed on both sides of the brake disc 9, the right plate being connected with piston 6, and the left plate being connected with a caliper. The disc 9 is fixed on and rotates with the wheel. The brake fluid in cylinder 5 is compressed when the plunger moves left. The piston *6* is driven to the left by compressed fluid and presses the disc to slow down the wheel.

As mentioned previously, the EMLA converts electrical energy into electromagnetic force, then the force is enlarged by the UDHC and acts on the brake disc directly. Therefore, the structure of the DDEHB is simpler than those of the EMB and EWB because a motion conversion mechanism such as a ball-screw mechanism is unnecessary in the DDEHB [19]. More importantly, the working mode of the traditional hydraulic brake, which is the typical 'master cylinder boost–hydraulic pressure transfer–wheel cylinder boost', is completely abandoned in the DDEHB. Therefore, the response of the DDEHB is faster than those of the EMB and the traditional hydraulic brake.

### 2.2. Model of the DDEHB

It is necessary to build accurate mathematical models for controller designing. The DDEHB actuator can be divided into three subsystems, which are electric subsystem, magnetic subsystem, and mechanical-hydraulic subsystem [20].

The electric subsystem can be equivalent to a closed circuit consisting of inductor, resistor, and counter electromotive force, which is described by Equations (1) and (2):

$$U_e = E_e + R_e i_e + L_e \dot{i}_e \tag{1}$$

$$E_e = B_e l_e N_e \cdot v_e = K_e v_e \tag{2}$$

where $U_e$ is the voltage of the EMLA, $i_e$ is the coil current, $R_e$ is the coil resistance, $L_e$ is the coil inductance, $E_e$ is the coil counter electromotive force, $B_e$ is the magnetic induction intensity, $l_e$ is the coil single-turn length, $N_e$ is the number of coil turns, and $v_e$ is the velocity.

If the magnetic hysteresis losses and end effects are ignored, the magnetic subsystem can be simplified as a proportional system:

$$F_e = B_e l_e N_e \cdot i_e = K_m i_e \tag{3}$$

where $F_e$ is the electro-magnetic force.

In order to simplify the mechanical hydraulic subsystem, unimportant factors such as liquid frictional drag and instantaneous shock are ignored. The simplified model can be described by Equations (4)–(7) according to the literature [21]:

$$F_e = m_e \dot{v}_e + C v_e + p S_l \tag{4}$$

$$S_l \dot{x}_l = S_i \dot{x}_i + 10^3 \times \frac{S_i x_i + V_0}{K_c} \frac{d(p - p_s)}{dt} \tag{5}$$

$$p_s = \frac{F_s}{S_i} \tag{6}$$

$$F_s = K_{ot} \cdot x_i \tag{7}$$

where $m_e$ is the total mass of coil and plunger, $C$ is the coil damping factor, $p$ is the hydraulic pressure, $S_l$ is the cross area of the plunger, $x_l$ is the plunger displacement, $S_i$ is the cross area of the piston, $x_i$ is the piston displacement, $K_c$ is the equivalent distortion modulus of the hydraulic fluid, $V_0$ is the initial

volume of the cylinder, $p_s$ is the equivalent pre-pressure of the cylinder, $F_s$ is the equivalent pre-force of the cylinder, $K_{ot}$ is the stiffness of the sealing ring.

In the electric subsystem, if the state variable is defined as

$$x = i_e$$

the input variable is defined as

$$u = U_e$$

The state equation can be described by Equation (8).

$$\begin{cases} \dot{x} = -\frac{R_e}{L_e}x - \frac{K_e}{L_e}v_e + \frac{u}{L_e} \\ y = x \end{cases} \tag{8}$$

The piston presses the disc firmly during braking, so the displacement of the piston is maximum, and the velocity is 0. If the state variables are defined as

$$\begin{bmatrix} x_1 \\ x_2 \end{bmatrix} = \begin{bmatrix} p \\ v_e \end{bmatrix}$$

the input variable is defined as

$$u = F_e$$

with

$$K_k = 10^3 \times \frac{S_i x_{imax} + V_0}{K_c}$$

the state equation of the mechanical-hydraulic subsystem can be described by Equation (9).

$$\begin{cases} \dot{x}_1 = \frac{S_l}{K_k}x_2 \\ \dot{x}_2 = -\frac{S_l}{m_e}x_1 - \frac{C}{m_e}x_2 + \frac{1}{m_e}u \\ y = x_1 \end{cases} \tag{9}$$

The state equations of each subsystem and their interrelations are shown in Figure 2. If the EMLA voltage is $U_e$, it generates the current $i_e$ in the coil. The coil will generate the electromagnetic force $F_e$, inducing the displacement of $x_l$. The force constant $K_m$ of the EMLA is affected by the displacement $x_l$. At last, the current $i_e$ changes because of the varied force constant $K_m$ and coil velocity $v_e$.

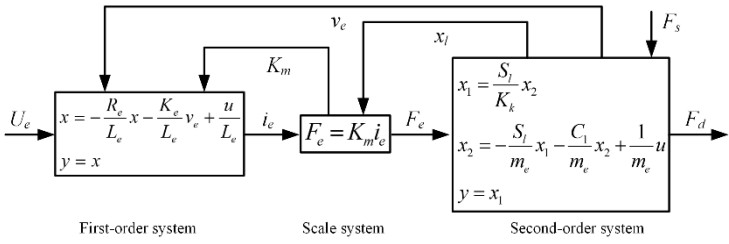

**Figure 2.** Mathematical model of the DDEHB.

## 3. Control Method of the DDEHB

As shown in Figure 2, the DDEHB consists of a first-order system, a proportional system, and a second-order system, in series. Therefore, a controller with double loop and anti-disturbance capability was designed according to the series structure, which is shown in Figure 3. The outer loop is the brake force (hydraulic pressure) controller, which calculates the coil current according to the target brake force (target pressure) and actual brake force (actual pressure); the inner loop is the current

controller, which calculates the voltage of the EMLA according to the target current calculated by the outer controller and actual coil current.

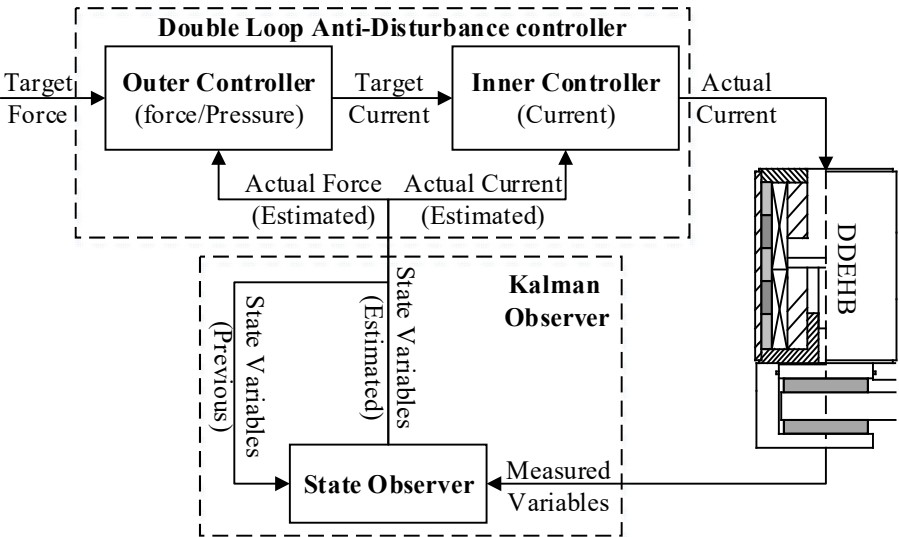

**Figure 3.** Structure of the DDEHB controller.

The feedback variables influence the controller. The measured variables usually contain process noise and measurement noise; however, there are often some variables that cannot be measured directly, such as the pressure variation ratio. Therefore, a state observer was designed to eliminate the noises and to estimate the pressure variation ratio.

### 3.1. State Observer

The state observer was based on a Kalman filter, which uses previous estimated variables and current measured variables to estimate the current state variables. Redefine the state variables on the basis of the Kalman filter as:

$$x = \begin{bmatrix} x_1 \\ x_2 \\ x_3 \end{bmatrix} = \begin{bmatrix} i_e \\ v_e \\ p \end{bmatrix}$$

If the input variable is defined as

$$u = U_e$$

the state equation of the DDEHB becomes

$$\dot{x} = A_c x + B_c u \tag{10}$$

where

$$A_c = \begin{bmatrix} -\frac{R_e}{L_e} & -\frac{K_e}{L_e} & 0 \\ \frac{K_m}{m_e} & -\frac{C}{m_e} & -\frac{S_l}{m_e} \\ 0 & \frac{S_l}{K_k} & 0 \end{bmatrix} \quad B_c = \begin{bmatrix} \frac{1}{L_e} \\ 0 \\ 0 \end{bmatrix}$$

The coil current and hydraulic pressure can be measured directly by sensors, so the measured variables are defined as:

$$z = \begin{bmatrix} z_1 \\ z_2 \end{bmatrix} = \begin{bmatrix} i_e \\ p \end{bmatrix}$$

Then, the measurement equation is

$$z = H_c x \tag{11}$$

where

$$H_c = \begin{bmatrix} 1 & 0 & 0 \\ 0 & 0 & 1 \end{bmatrix}$$

The Kalman filter consists of a series of recursive formulas which use the measured variable $z$ to estimate the state variables $x$ and effectively eliminate the process noise $\omega$ and measured noise $v$. The state Equation (10) and measurement Equation (11) are discretized as Equations (12) and (13) using the accurate discretization method:

$$x_k = Ax_{k-1} + Bu_{k-1} + \omega_{k-1} \tag{12}$$

$$z_k = Hx_k + v_k \tag{13}$$

where $k$ is the sampling time, and $A$, $B$, $H$ are defined by Equations (14)–(16), which are discretized state matrix, input matrix, and measurement matrix, respectively:

$$A = e^{A_c T} \tag{14}$$

$$B = B_c \int_0^T e^{A_c T} dt \tag{15}$$

$$H = H_c \tag{16}$$

where $T = 10^{-5}s$ is the sampling time.

Define $\hat{x}_k^-$ as the prior estimation state, $\hat{x}_k$ as the posterior estimation state modified by the measured variable $z_k$, $P_k^-$ as the prior estimation error covariance, $P_k$ as the posterior estimation error covariance, $K_k$ as Kalman gain, and $Q$ and $R$ as process noise and measure noise covariance matrix, respectively.

According to the Kalman principle, the priori estimation state of the $k$ step is

$$\hat{x}_k^- = A\hat{x}_{k-1} + Bu_{k-1} \tag{17}$$

The priori estimation error covariance at the k step is

$$P_k^- = AP_{k-1}A^T + Q \tag{18}$$

The Kalman gain at the $k$ step is

$$K_k = P_k^- H^T \left( HP_k^- H^T + R \right)^{-1} \tag{19}$$

The posteriori estimation state at the $k$ step is

$$\hat{x}_k = \hat{x}_k^- + K_k \left( z_k - H\hat{x}_k^- \right) \tag{20}$$

The posteriori estimation error covariance at the $k$ step is

$$P_k = (I - K_k H)P_k^- \tag{21}$$

The priori estimation state $\hat{x}_k^-$ at the $k$ step is calculated by Kalman recursive formulas according to Equation (17) and the posteriori estimation state $\hat{x}_{k-1}$ at the $k-1$ step. Then, the priori estimation state $\hat{x}_k^-$ is corrected as posteriori estimation state $\hat{x}_k$ according to the measured variable $z_k$ and Equation (20). $\hat{x}_k$ is able to effectively eliminate the measured noise and process noise and estimate the unmeasured variable (pressure variation ratio).

*3.2. Anti-Disturbance Controller*

3.2.1. Outer Controller

The design of the outer controller was divided into four steps, which were arrangement of the transition process, calculation of variable differences, calculation of the initial output, and disturbance compensation.

(1) Arrangement of the transition process.

There is usually a large difference between the actual braking force and its target value at the initial stage of braking or a sudden change of the target value. If the controller output is directly calculated from the difference between the actual and the target force, the actual braking force is easily overshot. Therefore, a transition progress is designed for the target force, and the actual force is required to track the transitional value so as to minimize or avoid the overshoot effectively. Han designed a transition process which is complex and difficult to apply to fixed-point DSP (Digital Signal Processor) (TMS320F2812) [22]. This work designed a similar transition process which is simpler and easier to be implemented on fixed-point DSP.

If the target hydraulic pressure is $p_o$, its transition value is $p_o^*$, and the differential of the transition value is $\dot{p}_o^*$, the transition process can be described as

$$p_o^*(k) = [p_o(k) - p_o^*(k-1)] \cdot r_e + p_o^*(k-1) \tag{22}$$

$$\dot{p}_o^*(k) = \frac{p_o^*(k) - p_o^*(k-1)}{h_e} \tag{23}$$

where $r_e$ is the tracking factor, and $h_e = 10^{-5}s$ is the sampling frequency.

(2) Calculation of the controller variable difference.

The state variable has been effectively estimated by the state observer:

$$\boldsymbol{x} = \hat{\boldsymbol{x}} = \begin{bmatrix} x_1 \\ x_2 \\ x_3 \end{bmatrix} = \begin{bmatrix} i_e \\ v_e \\ p \end{bmatrix}$$

So, the difference between the actual pressure and its transition value is

$$e_p = x_3 - p_o^* \tag{24}$$

The differential of the actual pressure can be calculated according to Equation (9):

$$\dot{p} = \frac{S_l}{K_k} v_e = \frac{S_l}{K_k} x_2 \tag{25}$$

So, the differential of the difference between the actual pressure and its transition value is

$$\dot{e}_p = \dot{p} - \dot{p}_o^* = \frac{S_l}{K_k} x_2 - \dot{p}_o^* \tag{26}$$

At last, the integral of the pressure difference can be calculated by

$$e_0 = \int_0^t e_p(\tau) d\tau \tag{27}$$

(3) Initial output of the controller.

Traditional PID controllers use a linear combination of difference, difference's differential, and difference's integral to calculate the output. However, practical experience shows that using a linear combination is not the best method. Therefore, a nonlinear combination was adopted, which was designed by Equation (28) [23]:

$$u_{o0} = k_0 fal(e_0, a_0, \delta_o) + k_1 fal(e_p, a_1, \delta_o) + k_2 fal(\dot{e}_p, a_2, \delta_o) \tag{28}$$

where $k_0$, $k_1$, and $k_2$ are the controller parameters, which are similar to the integral coefficient, proportional coefficient, and differential coefficient in the PID controller, and $fal(x, a, \delta)$ is the nonlinear function whose expression is

$$fal(x, a, \delta) = \begin{cases} \frac{x}{\delta^{(1-a)}} & |x| \leq \delta \\ sign(x)|x|^a & |x| > \delta \end{cases}$$

where the parameters $a$ and $\delta$ must satisfy the following conditions:

$$\begin{cases} a_0 < 0 < a_1 < 1 < a_2 \\ \delta_o = j \cdot h_e, j = 1, 2, 3 \cdots \end{cases}$$

The values of these parameters used in this study are shown in Table 1.

**Table 1.** Controller parameters.

| Parameter | $a_0$ | $a_1$ | $a_2$ | $\delta_o$ | $k_0$ | $k_1$ | $k_2$ |
|-----------|-------|-------|-------|------------|-------|-------|-------|
| Value | −0.73 | 0.69 | 3.74 | $3h_e$ | 6.3 | 3.4 | 0.04 |

(4) Disturbance compensation.

In the actual working process, some factors will affect the controller performance, such as coil temperature, frictional resistance, magnetic field end effect, liquid frictional drag, and instantaneous shock. Furthermore, there are other unpredictable disturbances in actual working conditions. All these factors were considered together as whole disturbance by Han [22]. In this work, the state observer was able to eliminate the process noise and measurement noise effectively but was not able to compensate the disturbance; therefore, the disturbance compensation method described in the literature [23] was adopted, which is expressed by Equations (29) and (30).

$$e = z_2 - x_3 \tag{29}$$

$$\dot{\gamma}_o = -\beta_o fal(e, 0.25, h_e) \tag{30}$$

where $\gamma_o$ is the whole disturbance, and $\beta_o$ is the perturbation parameter calculated by

$$\beta_o \approx \frac{1}{8.6 h_e^{2.2}} \tag{31}$$

Equation (30) can effectively estimate the total outer disturbance, so the final output of the outer controller after compensation is

$$u_o = \frac{u_{o0} - \gamma_o}{b_o} \tag{32}$$

where $b_o$ is determined by the system state equation, and its value is

$$b_o = \frac{K_m}{m_e} \tag{33}$$

### 3.2.2. Inner Controller

The transition process was designed in the outer controller, so it was unnecessary to redesign it in the inner controller, and the output $u_o$ of the outer controller was delivered directly to the inner controller as input variable. Also, the state variable $i_e$ of the inner loop was estimated simultaneously by the state observer, so the design of the inner controller was simpler. The inner controller was similar to the outer controller, which was described by Equations (34)–(41).

$$\text{Input variable}: \ i_i = u_o \tag{34}$$

$$\text{Controller variable error}: \ e_i = x_1 - i_i \tag{35}$$

$$\text{Initial output}: \ u_{i0} = k \cdot fal(e_i, a_i, \delta_i) \tag{36}$$

$$\text{Disturbance estimation}: \ e = z_1 - x_1 \tag{37}$$

$$\text{Disturbance estimation}: \ \dot{\gamma}_i = -\beta_i fal(e, 0.25, h_e) \tag{38}$$

$$\text{Disturbance estimation}: \ \beta_i \approx \frac{1}{1.6 h_e^{1.5}} \tag{39}$$

$$\text{Final output}: \ u_i = \frac{u_{i0} - \gamma_i}{b_i} \tag{40}$$

$$\text{Final output}: \ b_i = \frac{1}{L_e} \tag{41}$$

## 4. Simulation and Experimental Test

### 4.1. Simulation and Experimental Test of the DDEHB

In order to verify the feasibility of the DDEHB and the controller proposed in this paper, the ELMA was first manufactured and tested. The assembled ELMA and its main components are shown in Figure 4, and its main parameters are listed in Table 2. The electromagnetic properties were tested firstly. During the test, the ELMA voltage was fixed to 24 V, but the duty cycle of the H-bridge was changed from 0 to 100%, so the coil current also changed from zero to the maximum.

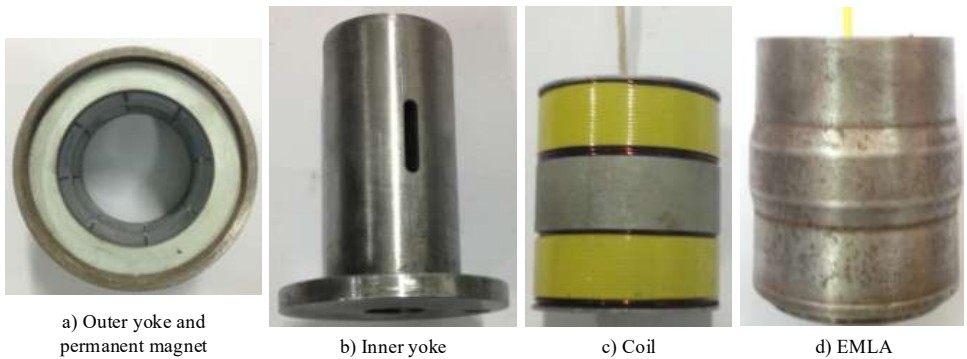

a) Outer yoke and
permanent magnet          b) Inner yoke          c) Coil          d) EMLA

**Figure 4.** Main components of the electro-magnetic linear actuator (ELMA) and assembled ELMA.

**Table 2.** Parameters of the EMLA.

| Parameter | Value |
|---|---|
| EMLA diameter/mm | 60 |
| EMLA length/mm | 70 |
| EMLA voltage/V | 24 |
| Coil resistance/Ω | 0.7615 |
| Coil inductance/μH | 279.8 |
| Peak current/A | 25 |

The force characteristics of the ELMA are shown in Figure 5. The output force is zero when the current is small because the electromagnetic force is insufficient to overcome the friction and it is proportional to the current once the electromagnetic force is greater than the friction. In the fitting curves, also shown in the figure, the primary coefficient is the ELMA force constant, and the constant term is the friction resistance. The force constant is slightly smaller if the coil is in the initial position and does not affect the DDEHB performance because the output force is only needed to overcome the movement resistance in this position. The force constant is very close to 15 when the coil is in the working position, thus the DDEHB is able to provide sufficient force with the help of the unequal-diameter hydraulic cylinder.

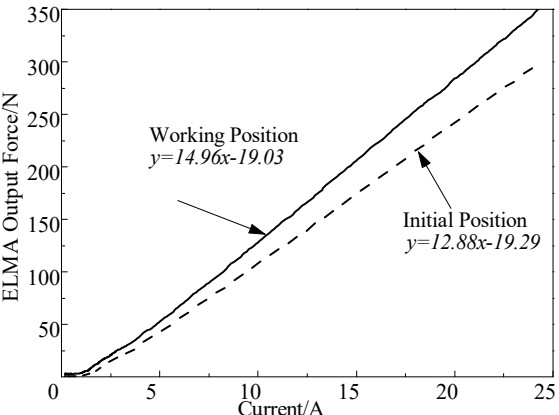

**Figure 5.** ELMA output characteristic.

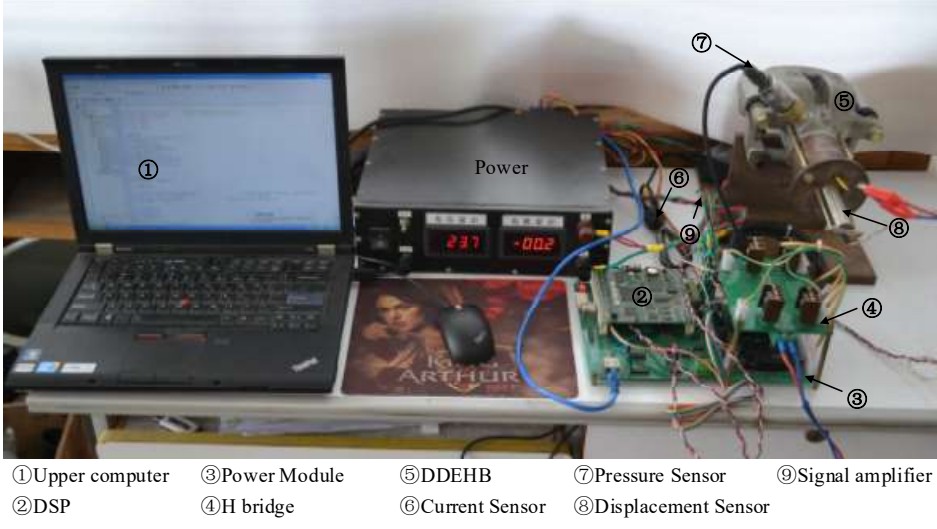

| ①Upper computer | ③Power Module | ⑤DDEHB | ⑦Pressure Sensor | ⑨Signal amplifier |
|---|---|---|---|---|
| ②DSP | ④H bridge | ⑥Current Sensor | ⑧Displacement Sensor | |

**Figure 6.** Prototype of the DDEHB and experimental platform.

After completing the ELMA test, a DDEHB co-simulation model and a prototype were established to verify the feasibility of the DDEHB and the controller described previously. The co-simulation model was based on Simulink and AMESim. The Simulink was used to establish the controller strategy including state observer and anti-disturbance controller, the AMESim was used to establish the DDEHB model including ELMA, unequal-diameter hydraulic cylinder, and brake caliper. The DDEHB prototype is shown in Figure 6, and its main parameters are listed in Table 3. The brake caliper was modified from a hydraulic unit and fastened to the ELMA by bolts; a digital signal processor TMS320F2812 (Texas Instruments, Dallas, TX, USA) was used as the controller chip, and a current sensor was used to measure the coil current. Due to space limitations for the caliper, it was not possible to measure the piston force directly, therefore a hydraulic sensor was used to measure the caliper force indirectly.

**Table 3.** Parameters of the DDEHB.

| Parameter | Value |
|---|---|
| Piston diameter/mm | 38 |
| Plunger diameter/mm | 6 |
| Max pressure/mPa | 10 |
| Max braking force/N | 10,343 ($\mu$ = 0.38) |
| Max piston pressure/N | 27219 |
| Electromagnetic force/N | 339 |

Firstly, the response characteristics were tested by simulation and experimental analysis. The target pressure was set to 5 mPa, and the results are shown in Figure 7. From the simulation curve, ADC experimental curve, and PID experiment curve, it can be seen that the DDEHB responded very quickly. Its response time was 15 ms, which is not only less than that of the conventional hydraulic brake, whose response time was 100 ms, but also less than that of the EMB, whose response time was 63 ms [24]. This shows that the DDEHB has a responsive advantage. Secondly, it can be seen that the state observer was able to reduce the measurement noise effectively by comparing the measured and the estimated data. Therefore, the ADC controller can improve control accuracy with smoother and more accurate feedback variables. Finally, the ADC controller was able to effectively suppress the overshoot thanks to the design of the transition process; in contrast, the PID controller does not have transition process and state observer, so its experiment curve showed obvious overshoot and significant fluctuations.

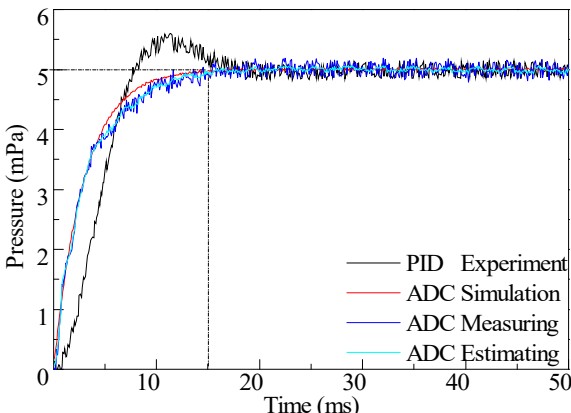

**Figure 7.** Step response of the DDEHB. PID: Proportion Integration Differentiation, ADC: anti-disturbance controller.

The anti-disturbance capability of the ADC was also tested, as shown in Figure 8. During the experiment, the pressure was kept stably at 5 mPa, but the measuring data were changed artificially to 6 mPa at 20th ms, so as to simulate the disturbance. As it can be seen from two curves, the ADC

responded more quickly to the disturbance and presented almost no flutter, while the PID controller responded more slowly and showed larger fluctuations. This indicates that the ADC could effectively improve the anti-disturbance ability of the DDEHB.

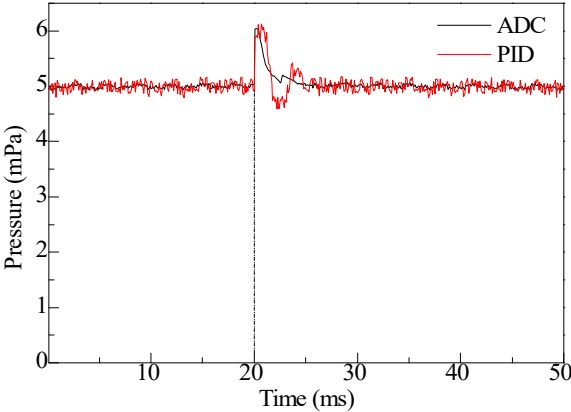

**Figure 8.** Anti-disturbance of the controller.

At last, a tracking experiment was designed to verify the pressure tracking capability. The target pressure was a sinusoidal periodic signal with maximum value of 4 mPa and cycle time of 1000 ms. The simulation and experimental results are shown in Figure 9, with a partial enlargement are shown in Figure 10. The pressure-tracking characteristics showed that the ADC was able to track the target value accurately. The experimental results agreed with the simulation results, indicating that the ADC was able to effectively control the hydraulic pressure or braking force. In addition, the high-pressure liquid might leak form the connection between EMLA and caliper, because the EMLA and the caliper were not an integral structure. However, the experimental results showed that the ADC was still able to control the pressure perfectly, indicating that it has a good anti-disturbance performance. For comparison, the tracking results of the PID controller are also shown in Figures 9 and 10. It can be seen from these figures that the anti-disturbance ability of the PID was obviously worse than that of the ADC and the target pressure could not be tracked accurately.

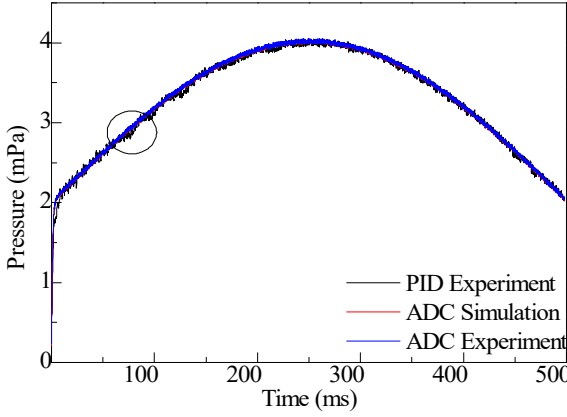

**Figure 9.** Tracking performance of the DDEHB.

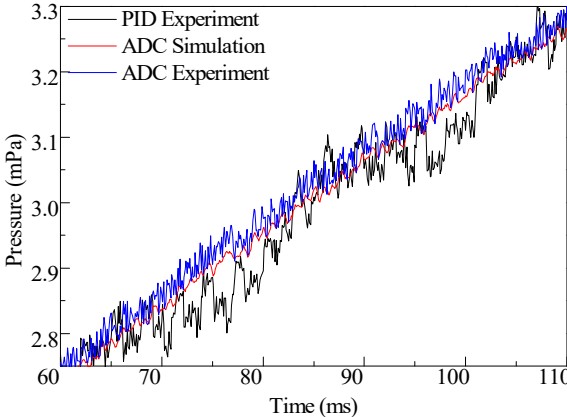

**Figure 10.** Tracking performance of the DDEHB.

*4.2. Simulation of Vehicle Performance*

The simulation and experimental results showed that the DDEHB is technically feasible and has the potential to improve a vehicle performance. Especially, the DDEHB more easily implements the anti-lock braking function than conventional hydraulic brake systems (conventional ABS). In order to research the anti-lock performance of a vehicle equipped with the DDEHB, co-simulation vehicle models with the DDEHB or the conventional hydraulic brake were established using Simulink and AMESim. The initial velocity was set to 30 m/s, the road maximum adhesion coefficient was set to 0.45, the target deceleration was set to 6 m/s$^2$, and the other parameters of the vehicle are listed in Table 4.

**Table 4.** Vehicle parameters.

| Parameter | Symbol/Unit | Value |
|---|---|---|
| Vehicle mass | M/kg | 1367 |
| Center of gravity | Hg/mm | 375 |
| Axle base | l/mm | 2400 |
| Front axle base | $l_f$/mm | 1056 |
| Rear axle base | $l_r$/mm | 1344 |

Shortly after the start of braking, both systems activated the anti-lock function. The velocity and deceleration are shown in Figure 11, and the wheel speed and slip rate are shown in Figure 12. The conventional ABS adopts the logic threshold control strategy, so the pressure in the wheel cylinder continues to circulate according to the "buck–keep–boost–keep" mode, and the wheel slip rates are maintained in a reasonable range (0.1–0.3) to prevent the wheels from locking. Therefore, it can be seen from the curves that the deceleration, wheel speed, and slip rate changed dramatically. In contrast, the DDEHB precisely controls the hydraulic pressure after identifying the road conditions, and the slip rates are maintained at their optimum values; therefore the deceleration, wheel speed, and slip rate are smoother than for the conventional ABS [25]. The final simulation results are listed in Table 5. We observed that the braking distance and braking time of the DDEHB were shortened by 12.19% and 15.54% compared with those of the conventional ABS, because the DDEHB allowed full road adhesion.

**Table 5.** Brake distance and brake time.

| Brake System | Brake Distance/m | Brake Time/s |
|---|---|---|
| Conventional ABS | 123.85 | 8.56 |
| DDEHB | 108.75 | 7.23 |
| Performance improvement | 12.19% | 15.54% |

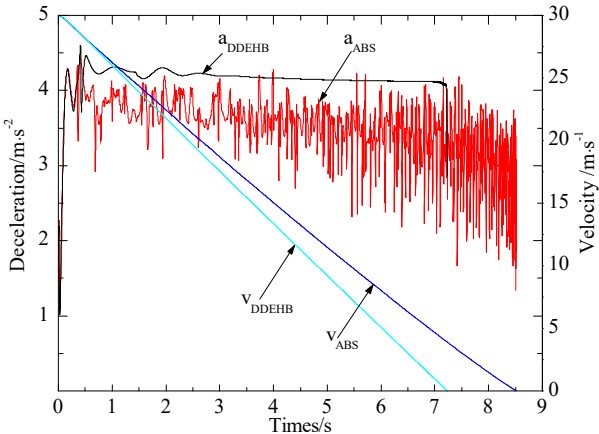

**Figure 11.** Velocity and deceleration.

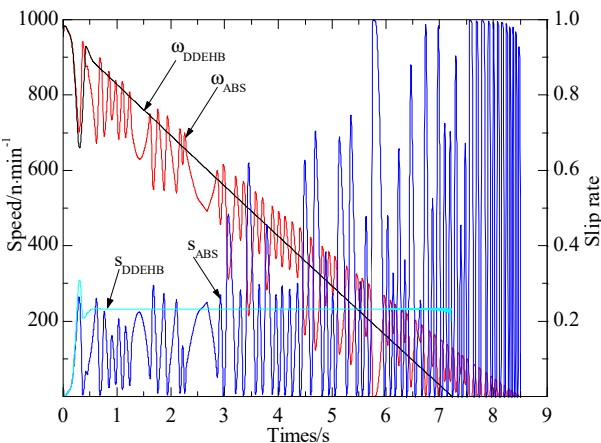

**Figure 12.** Speed and slip rate.

## 5. Conclusions

In this paper, a novel brake-by-wire actuator and its control method were studied. The simulation and experimental results showed that this actuator has good prospects. Its main advantages are described below.

(1) The DDEHB is able to control the brake force accurately, quickly, and independently, therefore, it is able to accurately control the braking process.

(2) The proposed state observer is able to eliminate the noise and estimate the state variables accurately, and the proposed anti-disturbance controller is able to accurately compensate the disturbance.

(3) The simulation results showed that the DDEHB is able to reduce the braking distance and the braking time significantly, in contrast to the conventional hydraulic brake system.

**Author Contributions:** Conceptualization, X.G.; formal analysis, L.Q.; investigation, L.W.; writing—original draft preparation, W.G.

**Funding:** This research was funded by the "Project of Science and Technology Research Program of Chongqing Education Commission of China, grant number, KJ1710240" and the "Project of Science and Technology Research Program of Chongqing, grant number, cstc2018jcyjAX0746".

**Conflicts of Interest:** The authors declare no conflict of interest.

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
