# Peer review of "Research on the Anti-Disturbance Control Method of Brake-by-Wire Unit for Electric Vehicles"

_wevj, doi:10.3390/wevj10020044_

Round 1
Reviewer 1 Report
In this work, the authors show a novel brake-by-wire unit based on the direct-drive electro-hydraulic system. The authors also propose a control method based on the disturbance rejection. The proposed control method demonstrates superior tracking performance compared to a classical PID. Moreover, it outperforms the classical rule based algorithm in case of ABS operation.
The reviewer suggests that the english is revised and the unit of measurement corrected (e.g. the multiplication factor 1000 is usually referred to as lowercase "k" and NOT capital "K"). Moreover, the following questions should be addressed before the final submission.
It is not clear to the reviewer which types of disturbances the authors mean to compensate and how this is analytically addressed. Please, elaborate this issue further in the text.
How is the conventional ABS algorithm being tested on the platform presented in the Fig 6. How does it compare to an industrial solution?
Author Response
Thanks the reviewers for the recommendations made on my paper, it is no doubt that these recommendations would help me to improve the quality of paper. So I will answer all the questions and comments made by reviewers one by one, and the modifications are also implemented in the paper.
Response to Reviewer 1 Comments
Point 1: The reviewer suggests that the English is revised and the unit of measurement corrected (e.g. the multiplication factor 1000 is usually referred to as lowercase "k" and NOT capital "K").
Response 1: The English sentence, units and format of the full text have been revised and corrected according to the recommendations. For details of the modification, please refer to the revised paper, which clearly indicates the trace of the revision.
Point 2: It is not clear to the reviewer which types of disturbances the authors mean to compensate and how this is analytically addressed. Please, elaborate this issue further in the text.
Response 2: There are many types of disturbances, including system disturbances and measurement disturbances. System disturbances mainly includes temperature change of coil, frictional resistance, end effect of magnetic field, loss of hydraulic path and instantaneous impact; measurement disturbances mainly includes temperature drift of sensor and measurement error of sensor. Due to the limitations of the author's knowledge, it is certainly impossible to list all the disturbances intact.
According to Jingqing's research [Jingqing, H., Auto Disturbances Rejection Control Technique. Frontier Science,2007. 1(1): p. 24-31.], the comprehensive results of various disturbances can be regarded as the summation disturbance, and it is estimated and compensated by equation (29)-(31) in the text, so that the complex disturbance compensation problem becomes a simple error feedback problem.
Point 3: How is the conventional ABS algorithm being tested on the platform presented in the Fig 6. How does it compare to an industrial solution?
Response 3: Figure 6 is a BBW actuator performance test platform, which can’t test the ABS algorithm of vehicle. The results of ABS in this paper are obtained by the co-simulation based on Simulink and AMESim, and the ABS algorithm is the logic threshold control strategy which is widely used in actual vehicles.
Reviewer 2 Report
see attachment

Author Response
Thanks the reviewers for the recommendations made on my paper, it is no doubt that these recommendations would help me to improve the quality of paper. So I will answer all the questions and comments made by reviewers one by one, and the modifications are also implemented in the paper.
Response to Reviewer 2 Comments
Point 1: In Kalman filter design, continuous time system (10) state equation is presented but same matrices are used in discrete-time model (12). This could not be used directly. Discretization method and sampling period must be specified. What is the initial value of covariance P0? What are the used values of process and measurement noise covariance matrices Q and R?
Response 1: Due to my lack of rigor and negligence that the paper is flawed, I would like to thank the reviewer for pointing them. The revised paper has modified according to the recommendation. The discretization method is the precise discretization method which has been described in the paper. The details of each matrix are as shown in the word file.
Point 2: If the disturbance is large and its model is not included in the Kalman filter, then the estimated states may not converge to true ones. How to solve this?
Response 2: In this paper, the Kalman filter is used to estimate the system state without compensating for disturbances. For the disturbance problem, the comprehensive effects of various disturbances are regarded as “summation disturbance” and compensated by equations (29)-(31) according to Jingqing's research [Jingqing, H., Auto Disturbances Rejection Control Technique. Frontier Science,2007. 1(1): p. 24-31.], so that the complex disturbance compensation problem is changed into simple error feedback problem.
Point 3: Sampling frequency he in (20) should be given.
Response 3: Sampling frequency he has been explained in the corresponding place in the text.
Point 4: All parameters of (25) and of fal(x,a,δ) should be detailed. How to select all of them in informed manner?
Response 4: The parameters of controller are listed in the following table and listed in the corresponding places in the paper. References [19] and [20] specify the range of each parameter. The parameters in the word table are selected after repeated trials and comparisons within the range.
Point 5: The selected design of anti-disturbance controller in double closed loop is not proper. The inner loop should be first designed, the outer one only afterwards. More comments and justification are needed.
Response 5: The inner loop controller is first designed during the actual design process. However, the design of anti-disturbance controller contains four steps, in which the first step is to design the transition process for the controlled variable. As the paper states, the transition process only needs to be designed for the final controlled variable (brake force/hydraulic pressure), and the intermediate variables do not need to be redesigned, therefore the transition process was designed only in the outer loop. In terms of paper structure, if the outer loop is first described, the design steps and ideas of the anti- disturbance controller can be fully explained.
Point 6: More recent works should enhance the literature review of intelligent braking systems solutions. A few ones to mention are: Data-driven model free slip control of anti-lock braking systems using reinforcement Q-learning, Sub-optimal switching in anti-lock brake systems using approximate dynamic programming, Design and Analysis of Output Feedback Constraint Control for Antilock Braking System with Time-Varying Slip Ratio.
Response 6: Some latest literatures have been added in the introduction, and I have read and referenced the recommended literatures.
Point 7: How was it obtained the slip rate of Fig. 12? Measurement or equation?
Response 7: The curve in Fig. 12 is obtained by co-simulation. The slip ratio is not measured directly, which is calculated by the feedback of the wheel speed, the longitudinal velocity and deceleration of vehicle.
Point 8: Some confusions and typing faults:-”k is simple time” vs. ”k is sample time”- Revise the statement ”So the nonlinear combination described Chen is adopted”- ”usually has noise such a process noise...”
Response 8: The English sentence and format of the full text have been revised and corrected according to the recommendations. For details of the modification, please refer to the revised paper, which clearly indicates the trace of the revision.

Round 2
Reviewer 2 Report
Many more details were added in the revised version about the scientific method. It is positive that the reviewer's recommendations were taken into account. The presentation is more transparent and its quality is improved.
A final proofreading is required.